# Chinese Public Response to Occupational Safety and Health Problems—A Study Based on Psychological Distance

**DOI:** 10.3390/ijerph16111944

**Published:** 2019-05-31

**Authors:** Shanshan Li, Hong Chen, Xinru Huang, Congmei Hou, Feiyu Chen

**Affiliations:** School of Management, China University of Mining and Technology, Xuzhou 221116, China; shanshanli0809@163.com (S.L.); hxr881003@163.com (X.H.); houcongmei333@163.com (C.H.); chenfeiyu@cumt.edu.cn (F.C.)

**Keywords:** occupational safety and health, psychological distance, public response, response gap

## Abstract

*Background:* The effective governance of occupational safety and health problems is inseparable from public participation and response. *Methods:* Based on the perspective of psychological distance, this paper adopted a quadratic response surface regression analysis method to investigate cognitive, emotional, expected and behavioral distances to occupational safety and health topics and their corresponding responses. *Results:* As demonstrated by the data statistics and response surface regression analysis results for 2386 valid samples, the relatively close psychological distance dimensions of the public with regard to occupational safety and health problems indicated the high endogenous tendency of the public to pay attention to occupational safety and health problems. The consistency between public cognitive and emotional distance with regard to occupational safety and health presented a “progressive decrease” in response towards behavioral distance, whereas the consistency between cognitive and expected distance reflected “convex” changes towards behavioral distance. Finally, the consistency between emotional and expected distance generally presented a “progressive increase” response towards behavioral distance. *Conclusions:* This research provides information regarding the public awareness of and response to occupational safety and health issues and how to promote occupational safety and health issues in order to improve them.

## 1. Introduction

The acceleration of global economic integration, liberalization of worldwide trade and investment in and application of new technologies, new materials, and new crafts in modern industry are causing the categories and number of occupational accidents and occupational diseases to increase continually in numerous developing countries. According to the statistics of the International Labor Organization (ILO), there are around 250 million production casualty accidents per year worldwide, with 475.6 accidents per minute on average. Among these, the death toll for production accidents and labor diseases totals about 1.1 million. In addition, the ILO estimates that by 2020, the incidence of labor diseases in the world will double and the global occupational safety and health situation will present an obvious tendency to deteriorate. The occupational safety and health situation in China will become increasingly austere—200 million workers are already currently suffering from the hazards of occupational diseases in varying degrees according to 2018 State Administration of Work Safety data. Statistics from the Ministry of Human Resources and Social Security indicate that the working-age (16–59) population in China exceeded 900 million in 2017 and this number may stabilize at around 800 million up to 2035. Occupational safety and health problems involve the core rights and interests of every worker. They not only affect the production activities in the working world, but also closely connect with the happiness of every family throughout society.

Occupational safety and health has typical attributes of public health concern. According to Spijkers and Honniball [1], public participation is conducive to the handling of public health concerns. As a result, the development of the occupational safety and health cause is inseparable from the public society, because public concern and support directly determine the social atmosphere which confronts occupational safety and health problems. However, the public tend to ignore occupational safety and health problems. In particular, in comparison with industrial accidents, occupational diseases and relevant symptoms are usually invisible to a large extent. In this sense, it is necessary to probe into public attitudes and the response to occupational safety and health problems. By introducing psychological distance, this research was systematically and comprehensively designed to investigate public awareness of and attitudes to occupational safety and health problems.

Occupational health psychology has been preceded by over a century of inquiry in psychology, sociology, philosophy, and other disciplines regarding the conditions of work and the welfare of workers, organizations, and society [2]. In 1912, Bullough proposed the concept of psychological distance, and its scope of application gradually expanded from aesthetic principles to the social group field of study for the measurement of social attitudes [3], and later to the trade sector for the description of individual perception [4] and to the domain of social relations [5]. With growing attention from researchers, studies of psychological distance have gradually but continually expanded to other fields. Agnew et al. defined psychological distance as “individuals’ subjective perception about others and self-relation distance and resulting emotional experience after integrating all sorts of social information” [6]. Trope and Liberman deemed psychological distance to be “individuals’ perception about different time contexts, space contexts, social relations and possible event occurrence contexts at this very moment” [7]. Both of these explanations regard psychological distance as a kind of perception, and a kind of psychological construction made by individuals based on the interpretation and processing of objective information. Psychological distance is used to explain people’s response mechanisms for object perception and assessment decisions. With regard to occupational safety and health problems, the public also perceive and make assessment decisions with a self-centered perspective. For this reason, it is appropriate to start from psychological distance in order to discuss public intimacy with occupational safety and health problems.

In combination with relevant knowledge of psychological distance, this research builds up a four-dimensional structure for occupational safety and health psychological distance, starting from the dimensions of cognitive, emotional, expected and behavioral distance to explore public attitudes and responses to occupational safety and health problems. Cognitive distance refers to public occupational safety and health knowledge status and degree of cognition; emotional distance refers to public emotional perception of the intimacy and degree of integration of occupational safety and health problems; expected distance refers to public perception of the future expectation of occupational safety and health problems based on existing tendencies or judgment experience; and behavioral distance refers to the public perception of participatory behavior in occupational safety and health governance. Against this backdrop, residents in six cities from eastern, middle and western regions of China were selected for investigation and acquiring information on public psychological distance intimacy with regard to occupational safety and health problems. Furthermore, by analyzing the relationships and mechanisms of influence among cognitive, emotional, expected and behavioral distance, our research was designed to provide information for the further study of occupational safety and health problems in China. This research is significant for the promotion of occupational safety and health governance.

## 2. Theoretical Analysis

### 2.1. Implication Analysis of Public Psychological Distance towards Occupational Safety and Health Problems 

Existing studies on occupational safety and health mostly concentrate on the structural aspects of the occupational safety and health system [8] and start from the perspective of state policy to prevent and solve occupational safety and health problems. Although policy efficacy itself plays a vital role in the solution of occupational safety and health problems, the public has a more important status as the executor of policy. On the other hand, occupational safety and health problems may also be governed from the perspective of occupational safety and health supervision. Relevant studies list the establishment of state occupational safety and health supervision agencies and the enforcement of effective supervision as an important move to ensure occupational safety and health [9]. It cannot be denied that supervision may be useful in urging practitioners to reduce unsafe actions and to normalize employee operation, but public attention on occupational safety and health problems is probably more persistent and stable because of endogenous factors (such as health, civic and responsibility awareness). As demonstrated by academic studies, the public have a stronger sense of identity towards garbage regulation policies as a result of endogenous factors [10,11]. Existing studies do not start from the public perspective in order to observe their attitudes and response towards occupational safety and health. Therefore, no solutions have been made for the governance of occupational safety and health in essence.

Distance in natural science generally means the interval length of objects in space or time, while psychological distance emphasizes the importance of individual perceptions and comprehension of the environment as a kind of social cognitive theory. One of the core tenets is that people’s responses towards social events depends upon their psychological representation [12]. Individuals’ psychological distance towards objects reflects their intimacy with, and emphasis on, the objects and directly determines individual behaviors. Consequently, this research incorporates psychological distance into the study of occupational safety and health problems. Psychological distance manifests individuals’ intrinsic perceptions about intimacy. In this sense, public psychological distance towards occupational safety and health explicitly reflects their concern with occupational safety and health.

In the study of psychological distance, different researchers give interpretations from different perspectives. From the perspective of trade, psychological distance is defined as the factor which hinders or disturbs suppliers and consumers [4]. From an interpersonal relations angle, psychological distance is defined as individual subjective feelings towards intimacy or alienation generated from the sense of uncertainty about surrounding relations, subject to the discrepancy in status, values and cultural backgrounds. In the field of organization management, psychological distance is defined as employees’ subjective judgments about distance intimacy, which predict and evaluate their behavior within the organization based on their degree of acceptance and actual degree of contribution. It is also used to describe the degree of conformity or integration between subjects and perception objects [13]. In all research fields, psychological distance in nature is a kind of subjective judgment. Likewise, in the field of public health, the public also makes subjective judgments about intimacy with or alienation from occupational safety and health. Accordingly, this research defines psychological distance as a subjective, public intimacy perception with a resulting promotion of action and inclination based on the comprehension and degree of perception of occupational safety and health problems.

### 2.2. Structural Analysis of Public Psychological Distance towards Occupational Safety and Health Problems 

At present, studies concerning psychological distance primarily exhibit the following four dimensions: spatial distance, temporal distance, social distance and hypotheticality. Spatial distance is defined as the distance between stimulant and individual in the spatial dimension; temporal distance is defined as the time between individual and present and target events in the past or future; social distance is defined as the intimacy or specificity of the relation between social objects and individual; hypotheticality is defined as the occurrence possibility of event, existence possibility of thing, or closeness degree to individual real life [7]. In their study of employee-organization psychological distance, Chen and Li grouped psychological distance into six dimensions: experiential, behavioral, emotional, cognitive, time-space and objective social distance [13]. For example, if employees felt close to organization time-space distance, they would like to stay in the organization (voluntary overtime). Otherwise, they might get out of the organization immediately after work. If employees felt close to organization emotional distance, they would have sense of happiness in the organization or otherwise suffer from pains and depression. If employees felt close to organization behavioral distance, they might sacrifice their own interests to safeguard the interests of the organization. On the contrary, they would pursue their own best interests in the organization. This research proposes that public perceptions regarding psychological distance towards occupational safety and health also vary at different levels.

As demonstrated by research, public perceptions about objects reveal their focus on those objects. Li recruited urban citizens as research objects and deemed public perception to be the comprehension and degree of perception of public and social affairs [14]. In a study on residents’ focus on health knowledge, Dai et al. observed that the health knowledge aspects that concerned individuals the most were, in order of importance, food safety, chronic disease, infectious disease and psychological health, and that they accordingly developed pertinent health promotion strategies [15]. Given that public perceptions of occupational safety and health embody public psychological distance intimacy in occupational safety and health, this research illustrates public cognitive distance towards occupational safety and health as a measurement dimension of psychological distance, and highlights the assessment of the degree of public comprehension of relevant occupational safety and health knowledge.

Emotions are individuals’ attitudes and experiences regarding whether objective things have satisfied personal demands, and include love, pleasure, happiness, detestation, anger, or contempt. Emotion is the immediate reflection of relation intimacy. As found by relevant studies, establishing psychological contacts can induce individuals to maintain similar emotional and even physiological states to others [16]. This implies that when individuals keep emotional intimacy with others (or other objects), their focus on those objects can be inferred to some extent. While studying the employee-organization relationship, Chen and Li used emotional distance to express employees’ emotional intimacy and integration perceptions about their organization [13]. In the same way, public perceptions of occupational safety and health problems also involve a certain degree of emotional integration. Such emotions may include a sense of depression or anger towards occupational safety and health problems that have occurred, or a sense of happiness and comfort resulting from the progress made with regard to existing occupational safety and health problems. Hence, this research chose public emotional distance towards occupational safety and health as a measurement dimension of psychological distance, and emphasized the measurement of public emotional integration in occupational safety and health problems.

The term “expectancy” refers to an estimation regarding future events. From the perspective of realistic behaviors, expectancy is a process whereby subjects make specific behavioral decisions pursuant to a judgment made with collected information about the future. At present, available studies about public expectancy mainly emphasize the macroeconomic expectancy management field. Bernanke held the opinion that currency policy is essentially about the problem of expectancy [17]. Public expectancy management has become the key content and core of the currency policy of central bank macro-control work [18]. To describe the employee-organization relationship, Chen and Li put forward the concept of experiential distance to represent employees’ perceptions about future expectancies based on existing tendencies or experiences [13]. From this view, the public would also form some explicit perception about the future direction of occupational safety and health on account of acquired information related to occupational safety and health problems. This is known as the expected distance.

Behavior includes all purposeful activities of organizations or individuals, namely the conduct and actions of subjects. Qin and Jing started from the perspective of behavioral distance to explore behavioral difficulties in management, and their study provided a new channel for investigation of the harmonious development of organizations [19]. Chen and Li put forward behavioral distance to represent employees’ perceptions about organizational intimacy in terms of “favor-organization” behavior [13]. Actions here specifically mean organizational citizenship behaviors, namely employees’ out-of-duty behaviors in favor of the organization, including personal initiative, helping behaviors and organization loyalty [20]. Similarly, this means that the public needs to have a relatively close psychological distance towards occupational safety and health in order to solve related issues. As a consequence, public behavioral willingness and responses towards occupational safety and health problems reflect their focus on occupational safety and health. Public behavioral distance towards occupational safety and health is a key indicator of psychological distance.

### 2.3. Structural Relation Analysis of Psychological Distance

Cognition determines the way that individuals perceive objects and has a significant influence on individual behavioral patterns. Researchers have conducted many studies on the relation between cognition and behavior. In the environmental studies field, Vringer et al. proposed that recognition of environmental problems and diverse insights about the role of environmental behaviors strongly influenced residents’ energy consumption behaviors [21]. According to Groot and Steg [22], environmental cognition, environmental knowledge and concern for the environment could significantly affect residents’ choice of travel mode. In education, Li and Liu explored the relation between left-behind middle school students’ cognitive emotion regulation and characteristics of dangerous behaviors, finding that passive cognitive emotion regulation was a risk factor for dangerous behaviors, while positive cognitive emotion regulation was a protective factor for dangerous behaviors [23]. Shao used the example of an emergency incident to probe into the rhetoric strategies and communication paths of news releases under the perspective of public cognition [24]. Given that this research concentrates on public familiarity with occupational safety and health-related knowledge with regard to public cognitive distance towards occupational safety and health, the relation between knowledge and behavior is also a key concern of our study. As demonstrated by research findings, knowledge has a significant impact on behavior, and related knowledge concerning specific behaviors is an important predictive variable for those specific behaviors [25]. Duerden and Witt also pointed out the significant correlation between environmental knowledge and individuals’ environment-related behaviors [26]. As a result, this research speculates that there exists a certain relationship between public cognitive distance towards occupational safety and health and behavioral distance.

Research has also shown the significant influence of emotion on behavior [27,28]. Specifically, in their study of green purchasing behaviors, Kanchanapibul et al. restricted influential factors to emotional variables, concluding that the standardization path coefficient of emotion against green purchasing behaviors was as high as 0.489 [29]. Wang and Wu observed that positive emotion had a stronger influence on green purchasing behaviors than negative emotion, through further exploration of the correlation and inherent laws of emotion and behavior [30]. Cheng et al. discussed the relation between college students’ cognition and procrastination behavior and eventually demonstrated a positive correlation between passive emotion and procrastination behavior, and a negative correlation between positive emotion and procrastination behavior [31]. Accordingly, this research tests the hypothesis that there exists a certain relationship between public emotional distance towards occupational safety and health and behavioral distance.

As a psychological phenomenon, expectancy is another key factor that affects individuals’ behaviors. Hussain stressed that changes in future information lead to changes in economic agents’ expectations and further manipulates their specific investment, consumption, labor supply and other behavioral decisions in the current period [32]. The study of public expectancy has been widely applied in all fields. In the field of consumption, Zhai analyzed multiple forms of expression of residents’ negative consumption expectancies, including a reduction in income, growing expenditure in the future, and a rise in product price [33]. Subsequently, pertinent measures were proposed to regulate these negative expectancies so as to promote consumption behaviors. Consumption expectancy had a direct influence on consumption demand as the prerequisite for consumers to formulate and implement consumption behaviors. In the domain of macroeconomics, Lu demonstrated the influence of public expectancy on the macroeconomy in a study of publically expected monetary policy effects [34]. Therefore, this research speculates that a certain relationship exists between public expected distance towards occupational safety and health and behavioral distance.

In summary, there exists certain relation among the four dimensions of public occupational safety and health psychological distance. Public occupational safety and health cognitive distance as the basis can adjust emotional distance and expected distance, and play a decisive role in behavioral distance. Furthermore, emotional distance can promote expected distance. If the public have closer occupational safety and health emotional distance, they probably have closer expected distance and vice versa. Simultaneously, emotional distance can directly trigger behavioral distance, which means that if the public have closer occupational safety and health emotional distance, they probably show high willingness of concern, willingness of implementing, willingness of dissemination and willingness of donations to occupational safety and health problems and vice versa. Expected distance can intensify behavioral distance. If the public have closer occupational safety and health expected distance, they probably have closer behavioral distance. The analysis of public psychological distance structural relationships with regard to occupational safety and health behaviors is shown in Figure 1.

## 3. Research Method

### 3.1. Quadratic Response Surface Regression Analysis

Based on our literature review, there exists a certain relation between the four dimensions in public occupational safety and health psychological distance (Figure 1), but until now few scholars have studied the influence of matching degree between cognition and emotion, cognition and expectation, emotion and expectation on behaviors. Public intimacy with occupational safety and health behavioral distance should not be simply ascribed to single factor like cognitive distance, emotional distance, behavioral distance, but should be the joint effect caused by the interaction of cognitive, emotional and expected distances. Therefore, discussion about the influence of matching degree between cognitive, emotional and expected distances on behavioral distance has very important theoretical and practical values. This research plans to conduct the research with quadratic response surface regression analysis method.

Response surface methodology utilizes integrated experimental technologies in statistics to solve the correlation between complicated system input (variable) and output (response). Mathematical expression of response methodology involves multiple linear regression (MLR) analysis, as shown in Equation (1):(1)$$y=\alpha_{0}+\sum_{j=1}^{n}\alpha_{j}x_{j}+\sum_{j-n+1}^{2n}\alpha_{j}x_{j-k}^{2}+\sum_{i=1}^{n-1}\sum_{j=i+1}^{n}a_{ij}x_{i}x_{j}$$
whereby *n* is the design variable, $\alpha_{0}$ is a constant term for an undetermined coefficient, $\alpha_{j}$ is a one-degree term for an undetermined coefficient and $\alpha_{ij}$ is a quadratic term for an undetermined coefficient.

Quadratic response surface regression analysis combines quadratic polynomial regression with response surface methodology. This method has been primarily applied in the study of individual-organization matching theory [35,36]. Subsequently, Chen et al. applied this method in the field of urban household garbage regulation policy to investigate the relationships between the degree of public policy understanding, support willingness, performance willingness, and promotion willingness [37]. In matching measurement research, there often include two measurement methods. The first one is direct matching measurement method which requires the respondent to directly determine the degree of matching between the two [38]. Though the method is rather simple, but the independent effects cannot be examined. Another method is indirect matching measurement which requests the respondent to respectively assess the features of the two, and then compare the consistency or similarity of their grading scores [39]. Indirect measurement usually adopts difference scores, Q classification method, product term indicator method, etc. However, these indirect matching measurement statistical strategies are defective in theory and methodology. For instance, difference scores can result in decreased measurement reliability and mutual confusion between the two effects [40]. Quadratic response surface regression analysis method belongs to a matching measurement and statistical analysis strategy developed to overcome above-mentioned defects [35]. It possesses the following advantages: Firstly, it provides an entire statistical analysis and explanation framework for matching measurement research that can not only well explain the secondary coefficient of polynomial regression equation, but also test the response surface features and shape formed by these coefficients. Secondly, it is not only limited to measuring the matching relation of representatives, but also can examine the influence caused by mismatching relation, thus providing a more robust evaluation method for the theoretical model related to matching research. Thirdly, compared with the traditional indirect matching method, it has higher predictive validity. Admittedly, the method also has limitations. Firstly, as quadratic response surface regression analysis adopts a series of statistical tests to judge the slope and curvature significance of all target lines on the response surface, it greatly increases statistical mistakes due to repeated examination [35]. Secondly, quadratic response surface regression analysis has the same problem with general regression analysis application. All of these methods assume measured indicator variables have no error, but this is not up to the real measurement conditions. Though quadratic response surface regression analysis does not totally jump out of traditional statistical analysis framework, available statistical analysis technology indicates that quadratic response surface regression analysis has extensive application prospects in matching or consistency research.

In our study, there was also a relationship between cognitive, emotional, expected and behavioral distance, which indicates a degree of “consistency” among all of these public distances towards occupational safety and health. This accords with the “matching” implied in individual-organization matching theory. Consequently, this research used quadratic response surface regression analysis to thoroughly investigate the response relationship between public cognitive, emotional, expected and behavioral distance in occupational safety and health. It defines consistency as the situation in which the score of cognitive distance equals that of emotional distance, cognitive distance equals expected distance, and emotional distance equals expected distance. The score is divided into five points according to a scale grade. Higher grades indicate a higher degree of consistency. In addition, this research calculates inconsistency (gaps) by comparing in pairs from the score of cognitive, emotional and expected distance, and separately calculating difference value in each pair.

### 3.2. Scale and Investigation

There is a lack of a mature scale for the study of public occupational safety and health psychological distance. Therefore, this research included an exploratory qualitative study to determine scale questions and collected material during interviews with 67 citizens associated with different industries, and of different age groups and characteristics. The interview material was sorted in strict accordance with open coding, axial coding and selective coding. In addition, six professionals (two professors, two associate professors and two lecturers) were invited to negotiate and determine question accuracy, scale feasibility and expression readability. Eventually, 16 questions were developed for the measurement of occupational safety and health psychological distance, on the dimensions of cognitive, emotional, expected and behavioral distance (see Table A1). Among these dimensions, cognitive distance primarily includes public cognition of occupational safety and health implications, the present situation, importance and occupational diseases. Emotional distance highlights the focus on occupational safety and health accidents, or state and corporate occupational safety and health problems, and public emotions such as grief, indignation and comfort. Expected distance is concerned with whether public perceptions of occupational safety accidents and the extent of injury are exaggerated, whether the quality of occupational safety and health problem governance has any influence on the future perceptions of the public, and whether the development of medical treatments and techniques ensures that occupational safety and health problems will not severely threaten the public. Behavioral distance deals with public understanding and willingness to concentrate on occupational safety and health problems, to commit to occupational safety and health problem governance, to popularize occupational safety and health knowledge to surrounding people, and to make donations to those injured in occupational accidents and occupational disease patients. Responses were on a Lickert five-point scale whereby 1 signified “totally incongruent”, 2 meant “not too congruent”, 3 meant “noncommittal”, 4 meant “relatively congruent”, and 5 meant “totally congruent”. A higher score indicated greater occupational safety and health psychological distance.

The pilot investigation took place in Jiangsu and Anhui Provinces in March 2018. Altogether 550 questionnaires for anonymous completion were distributed and 478 completed questionnaires were collected, with an effective response rate of 86.91%. As indicated by the descriptive statistical analysis of the test samples, the gender ratio of the samples was basically balanced, with males accounting for 59.7% and females accounting for 40.3%. Age distribution was relatively even, with participants aged below 20 years accounting for 1.3%, those aged 21–30 accounting for 25.1%, those aged 31–40 accounting for 29.6%, those aged 41–50 accounting for 23.4%, those aged 50–60 accounting for 14.8% and those aged above 60 accounting for 5.8%. The investigation targeted a variety of participants including government staff, coal mine enterprise leaders, and common social groups. The questionnaire was finally designed after reliability and validity analysis and correction of questions.

The formal investigation was composed of two parts. The first part collected individual socio-demographic information and the second part investigated public occupational safety and health psychological distance. The formal investigation started in April–July 2018. Altogether 3000 questionnaires were distributed and 2386 completed questionnaires were collected, with an effective response rate of 79.53%. Before distributing the questionnaires, regional respondents were selected via stratified sampling. Due to the diverse economic and regional features in the eastern, middle and western regions, the research screened two cities from each of the three regions: Hebei and Jiangsu Province in the eastern region, Anhui and Hunan Province in the middle region, and Sichuan and Xinjiang Province in the western region. Simultaneously, in consideration of the wide variety of potential respondents concerned with occupational safety and health problems, including government staff, coal mine enterprise leaders, coal mine safety supervisors, coal mine front-line workers, third-party social organization staff (including those working for occupational disease hospitals, occupational disease relief funds, and industrial association organization staffs), and patients with diseases such as pneumoconiosis as well as ordinary people, seven types of respondents were selected for the investigation to provide a sample including people from all walks of life and of both genders and different educational backgrounds, ages, and marital and political status. To be specific, the research chose three samples from 45 state-owned large and medium coal mine enterprises in China, and selected six types of coal mine front-line workers in ventilation and fire prevention, coal mining, tunneling, mechatronics, transportation, and ground work and leaders with different positions. Moreover, the research chose coal mine safety supervisors according to age and position discrepancy. For ensuring the rationality of third-party social organization staff samples, the research chose staff and volunteers from Chinese Occupational Safety and Health Association, China Coal Miner Pneumoconiosis Prevention Foundation, and Love Save Pneumoconiosis. The sample selection conformed to the practical distribution situation. The subject and purpose of the investigation were explained in detail to the participants. The researchers sent gifts to participants to express their gratitude and to improve the response rate and effectiveness of the questionnaire survey. Please refer to Table 1 for the specific sample structure.

### 3.3. Ethical Approval

This study was carried out in accordance with the recommendations of the Ethical Codes of Consulting and Clinical Psychology of Chinese Psychological Society, Chinese Psychological Society. The protocol was approved by the China Occupational Safety and Health Association—Occupational Mental Health Professional Committee. All subjects gave written informed consent in accordance with the Declaration of Helsinki. It is the duty of researchers who are involved in psychological research to protect the life, health, dignity, integrity, right to self-determination, privacy, and confidentiality of personal information of the research subjects. The responsibility for the protection of research subjects always rested with our research team and the China Occupational Safety and Health Association—Occupational Mental Health Professional Committee, and never with the research subjects, even though they had given consent.

## 4. Data Analysis

### 4.1. Factor Analysis

Formal questionnaire validity and reliability test results indicated that the Cronbach’s α value for public psychological distance towards occupational safety and health, at 0.830, was above 0.8. This implied that the scale had relatively high reliability in general. The Cronbach’s α values of all latent variables were 0.743, 0.661, 0.702 and 0.823 respectively and the corresponding CR values were 0.859, 0.805, 0.825 and 0.871. Given that both statistics exceeded the acceptability criteria, the scale passed the reliability test.

We then rigorously developed the program in line with the scale. Based on considerable literature research, the scale finally exhibited a favorable content validity after being repeatedly negotiated and corrected by five experts in the field of management. Moreover, the standardization load of 16 scale questions with corresponding latent variables, ranging between 0.584 and 0.815, was far above 0.5 and attained significance. The corresponding average variance extracted (AVE) were 0.604, 0.509, 0.545 and 0.629 respectively. Given the criterion of AVE > 0.5, the scale had good convergent validity. In addition, the AVE square root of the latent variables being above the correlation coefficient of the latent variables implied a favorable degree of latent variable structural discrimination. The scale accordingly passed the validity test.

AMOS 17.0 (SPSS Inc., Chicago, IL, USA) was adopted to carry out confirmatory factor analysis on the structural validity of the questionnaire. After two attempts at model regulation, the final fitting indicators were chi-square (χ^2^) = 3055, degree of freedom (df) = 887, χ^2^/df = 3.444, root mean square error of approximation (RMSEA) = 0.047, goodness of fit index (GFI) = 0.903, normed fit index (NFI) = 0.906, comparative fit index (CFI) = 0.915, incremental fit index (IFI) = 0.904, and Tacker-Lewis index (TLI) = 0.898. All indicators attained an ideal scope. In summary, the research verified the four-dimensional structure of public occupational safety and health psychological distance comprising cognitive, emotional, expected and behavioral distance.

### 4.2. Statistical Analysis

Descriptive statistical analysis was conducted for public occupational safety and health psychological distance and its four corresponding dimensions. Given that closer psychological distance indicates a higher public focus on occupational safety and health problems, the overall questions were marked in reverse. Lower scores showed closer public occupational safety and health psychological distance, while higher scores indicated farther public occupational safety and health psychological distance. (1–2) points indicated intimate distance, (2–3) points indicated relatively close distance, (3–4) points indicated relatively far distance and (4–5) points indicated far distance. The specific analysis results are shown in Table 2.

Table 2 shows that the mean value of public occupational safety and health psychological distance was 2.552. Over 60% of respondents and nearly 20% of respondents respectively showed a relatively close and intimate psychological distance to occupational safety and health. This implies that the public pays great attention to occupational safety and health problems. Specific analysis of these dimensions revealed that around 50% of respondents exhibited relatively far cognitive distance for occupational safety and health problems (in the range of 3–4 points), and that most respondents’ emotional, expected and behavioral distance towards occupational safety and health problems was relatively close, in the range of 1–3 points.

### 4.3. Quadratic Response Surface Regression Analysis

Analysis of the four dimensions of occupational safety and health, as shown in Figure 2, demonstrated that public cognitive, emotional, expected and behavioral distance all exhibit a gap. The group that had congruent cognitive, emotional, expected and behavioral distance (≥3 or <3) accounted for only 18.75%, while the group that was incongruent (i.e., had gaps) accounted for 81.25%. Additionally, the group with the scores of cognitive distance (≥3), emotional distance (<3), expected distance (<3) and behavioral distance (<3) accounted for 26.51% (highest occupation). In contrast, the group with the scores of cognitive distance (<3), emotional distance (≥3), expected distance (≥3) and behavioral distance (≥3) accounted for 0.17% (lowest occupation).

In combination with the above analysis, quadratic response surface regression analysis was used in a conceptual model to investigate the gaps in cognitive, emotional, expected and behavioral distance. From the perspective of behavioral distance, the following three models were constructed: cognitive distance/emotional distance model 1 (equations (2) and (3)), cognitive distance/expected distance model 2 (Equations (4) and (5)), and emotional distance/expected distance model 3 (equations (6) and (7)). In order to prevent multi-collinearity, centralization processing was applied to public cognitive distance (X), emotional distance (Y), expected distance (Z) and behavioral distance (M) towards occupational safety and health, and subsequently the quadratic components of X, Y and Z (X2, Y2, Z2) and the product terms X × Y, X × Z and Y × Z were calculated. The calculations were then completed using the SPSS 21.0 statistical software (International Business Machines Corporation, New York, NY, USA).
(2)$$\mathrm{Cognitive}/\mathrm{emotional}\;\mathrm{distance}\;M1:\;M1=\alpha_{0}+\alpha_{1}x+\alpha_{2}y+e$$
(3)$$\mathrm{Cognitive}/\mathrm{emotional}\;\mathrm{distance}\;M2:\;M2=\alpha_{0}+\alpha_{1}x+\alpha_{2}y+\alpha_{3}x^{2}+\alpha_{4}y^{2}+\alpha_{5}x\cdot y+e$$
(4)$$\mathrm{Cognitive}/\mathrm{expected}\;\mathrm{distance}\;M1:\;M1=\alpha_{0}+\alpha_{1}x+\alpha_{2}z+e$$
(5)$$\mathrm{Cognitive}/\mathrm{expected}\;\mathrm{distance}\;M2:\;M2=\alpha_{0}+\alpha_{1}x+\alpha_{2}z+\alpha_{3}x^{2}+\alpha_{4}z^{2}+\alpha_{5}x\cdot z+e$$
(6)$$\mathrm{Emotional}/\mathrm{expected}\;\mathrm{distance}\;M1:\;M1=\alpha_{0}+\alpha_{1}y+\alpha_{2}z+e$$
(7)$$\mathrm{Emotional}/\mathrm{expected}\;\mathrm{distance}\;M2:\;M2=\alpha_{0}+\alpha_{1}y+\alpha_{2}z+\alpha_{3}y^{2}+\alpha_{4}z^{2}+\alpha_{5}y\cdot z+e$$

Correlation analysis (Table 3) demonstrated the significantly positive correlation among public cognitive, emotional, expected and behavioral distance towards occupational safety and health problems. In further regression analysis results (Table 4), both behavior M1 and behavior M2 showed significant linear correlation with cognitive, emotional, expected and behavioral distance. Moreover, the regulation R2 of three-group behavior M2’s comparative behavior M1 was on a rising trend, which showed that behavior M2, with relatively strong explanatory power, was a more accurate representation of the correlation between the independent and dependent variable.

Edward points out the necessity to draw up a three-dimensional diagram to express the relation between independent variables and the dependent variable in case of any significance in a polynomial regression model with multiple quadratic terms. Consequently, response surface analysis was conducted on the regression model. The MatLab R2008a software (MathWorks, Natick, MA, USA) was used for programming and presentation of the three-dimensional diagram of public cognitive, emotional, expected and behavioral distance towards occupational safety and health. The Y = X transverse line indicates the line of congruence. On the X–Y plane, the two measurement indicators have equal value and orientation. A larger abscissa indicates higher congruence between the two indicators, while a larger ordinate indicates farther behavioral distance. To summarize, the response diagram of public cognitive, emotional, expected and behavioral distance towards occupational safety and health follows the Y = X transverse line (Figure 3b, Figure 4b and Figure 5b).

Next, the public response towards occupational safety and health cognitive, emotional and behavioral distance were analyzed. As suggested by the regression analysis results (Table 4), public emotional distance towards occupational safety and health problems in behavior M1 could significantly predict behavioral distance, but cognitive distance did not significantly predict behavioral distance. Public cognitive and emotional distance towards occupational safety and health problems in behavior M2 could significantly predict behavioral distance. The rising trend of behavior M2’s comparative behavior to M1 regulation R2 demonstrates the relatively strong exploratory power of behavior M2, and reflects the non-linear relation between independent variables and the dependent variable. To exhibit directly the response of public cognitive distance and emotional distance in occupational safety and health problems to behavioral distance and the influence of cognitive and emotional distance congruence (or incongruence) on behavioral distance, response surface analysis was carried out on the three-dimensional diagram of the regression model. Figure 3b reveals the “progressively decreasing” response of cognitive and emotional distance congruence towards behavioral distance, which means that higher congruence between cognitive and emotional distance leads to closer public behavioral distance towards occupational safety and health. In other words, a larger gap between cognitive and emotional distance indicates farther behavioral distance of the public towards occupational safety and health.

A similar method was adopted to analyze the response of public cognitive distance and expected distance towards behavioral distance in occupational safety and health. The regression analysis results indicated (Table 4) that public cognitive and expected distance towards occupational safety and health problems in behavior M2 could significantly predict behavioral distance. The significant rising trend of behavior M2’s comparative behavior to M1 regulation R2 demonstrated the relatively strong exploratory power of behavior M2 and reflected the non-linear relation between independent variables and the dependent variable. Figure 4b shows that the response surface congruence line Y = X “convex” throughout the response surface analysis on the three-dimensional diagram of the regression model. This implies that when cognitive and expected distance become congruent, public behavioral distance towards occupational safety and health first increases and later declines. The ultimate trend signaled that a smaller gap between cognitive and expected distance for public occupational safety and health leads to closer behavioral distance. The calculation result derived an inflection point coordinate on the response surface of (0.53, 0.38).

A similar method was also applied to the analysis of the response of public emotional distance and expected distance towards behavioral distance in occupational safety and health. The regression analysis results indicated (Table 4) that public cognitive distance and expected distance towards occupational safety and health problems in behavior M2 could significantly predict behavioral distance. The rising trend of behavior M2’s comparative behavior to M1 regulation R2 demonstrated the relatively strong exploratory power of behavior M2 and reflected the non-linear relation between independent variables and the dependent variable. Throughout the response surface analysis on the three-dimensional diagram of the regression model, Figure 5b displays the congruence calculation formula for emotional and expected distance. The congruence line Y = X shows that when cognitive and expected distance become congruent, public behavioral distance towards occupational safety and health problems first increases and later declines, and the general incongruence between emotional and expected distance presented the negative correlation with behavioral distance. In other words, although behavioral distance is at a relatively low level when cognitive and expected distance become incongruent, the final trend signals that a very small gap between cognitive and expected distance in public occupational safety and health perceptions leads to close behavioral distance. The calculation shows that the inflection point coordinate on the response surface was (0.76, 0.39).

## 5. Discussion

The descriptive statistical analysis of the sample demonstrated that approximately 80% of respondents had intimate or relatively close psychological distance towards occupational safety and health. Until 2017, the number of all types of workers in China totaled 776 million (National Bureau of Statistics of China 2017 data). Occupational safety and health problems concerned with the core rights and interests of workers are closely related to the survival and development of every single worker and their family. In this sense, public psychological distance towards occupational safety and health is relatively close overall. However, 62.44% of respondents presented a relatively remote cognitive distance towards occupational safety and health, or perhaps had insufficient understanding or knowledge of occupational safety and health. The research findings imply that this phenomenon is primarily attributable to two causes. First, the government or enterprise does not emphasize the publicity of occupational safety and health-related knowledge. Although they may have formulated many policies or regulations concerning occupational safety and health, inadequate publicity will result in the relatively far cognitive distance of the public. Second, the public lacks willingness to learn about occupational safety and health-related matters. Research shows that individuals encounter various barriers in the knowledge learning process [41,42]. Alternatively, individuals generally may not actively want to acquire such knowledge. Consequently, the government and relevant enterprises should optimize occupational safety and health-related knowledge publicity and overcome learning obstacles to the maximum extent possible.

The research demonstrated a gap among public cognitive, emotional, expected and behavioral distance towards occupational safety and health. This can be divided into two situations: a “weak attitude–strong behavior” situation (typical example: cognitive distance ≥3, emotional distance ≥3, expected distance ≥ 3 and behavioral distance < 3) and a “strong attitude–weak behavior” situation (typical example: cognitive distance < 3, emotional distance < 3, expected distance < 3 and behavioral distance ≥ 3). Research suggests that the cause of the “weak attitude–strong behavior” mismatch might be ascribed to an individual ability [43] and situational dilemma [44], with the former mainly including economic [45] and time restrictions [46]. However, in practice, there may be all kinds of restrictive situations in which the public cannot manage behavioral processing of public occupational safety and health issues. The cause of “strong attitude–weak behavior” is probably related to individuals’ self-presentation behaviors in the process of solving occupational safety and health problems. For instance, based on competitive altruism theory, Griskevicius et al. proposed that individual altruistic behaviors could be guided by stimulating individual momentum to seek status and identity [47]. Antonetti’s research also showed that the purpose of conspicuous green consumption was to pursue social status and reputation, and that it was also a type of altruistic behavior [48]. All of these are possible causes of the gap among cognitive, emotional, expected and behavioral distance.

The quadratic response surface regression analysis demonstrated that higher congruence between cognitive and emotional distance led to closer public behavioral distance towards occupational safety and health, which meant that congruence between of cognitive and emotional distance had a “driving effect” on behavioral distance. The Cognitive-Affective Personality System (CAPS) states that individual stable personality system structure is constituted by cognitive-affective units (CAUs) in some stable organizational relationship which may reflect individual peculiarities of personality. In addition, the events encountered by individuals interact with complex CAUs in the personality system and eventually determine an individual’s behavior [49]. Existing studies generally verify the close connection between personal internal congruence and behavior [50]. This also means that, on account of the understanding and relevant knowledge of occupational safety and health, the public will have much closer behavioral distance under emotional support.

It has been shown that congruence between public cognitive and emotional distance towards occupational safety and health and behavioral distance presents “convex” changes. When cognitive and expected distance become congruent, public behavioral distance towards occupational safety and health will first increase and later decline. When cognitive and expected distance become incongruent, two situations will emerge, namely “positive sequence” incongruence and “reversed sequence” incongruence. In “positive sequence” incongruence, the public still have noncommittal attitudes towards the expected effects of future occupational safety and health governance and this will further positively influence their behavioral distance. In “reversed sequence” incongruence, although the public has great confidence in future occupational safety and health governance, their expectations might be blindly optimistic. Expectancy based on insufficient cognition is unstable and lacking basis. Therefore, the behavioral distance is naturally very great. However, when the congruence between public cognitive and expected distance reaches a high level, public behavioral distance towards occupational safety and health will show a declining trend. In a study of cognition and expectancy, Zhang and Zhao also suggested that the interactive function among different cognitive biases that takes expectancy as the medium is the underlying cause which affects individual risk assets pricing [51].

It is worth noticing that the congruence between public cognitive and emotional distance towards occupational safety and health presents a “progressive increase” response towards behavioral distance. This phenomenon deviates from psychology and behavior. In combination with the congruence between public cognitive and expected distance, behavioral distance should also increase. Accordingly, it can be predicted that expectancy has a hysteresis effect [52]. In management psychology, a hysteresis effect is used to describe the phenomenon in which decision, management and many other factors fall behind situation development and trigger adverse consequences [53]. Expectancy is a kind of psychological representation. Expectancy hysteresis results in the primary rising and subsequent declining trend of behavioral distance, even though public cognitive and emotional distance towards occupational safety and health become closer. Simultaneously, the deviation between the congruence of public emotional and expected distance towards occupational safety and health and behavioral distance is a phenomenon of “incongruent speech and behavior”. The fundamental reason behind this may be the influence of a social commendatory effect [54]. When catering for social norms or management of impressions, the self-report bias tends to manifest itself in the form of positive attitudes (shown here as positive emotions and positive expectancy) and further deviate from following practical behaviors [55].

## 6. Conclusions

### 6.1. Conclusions

This study describes the connotations of the psychological distance of occupational safety and health for the public. Through literature and qualitative analysis, it has been shown that the psychological distance of the public with regard to occupational safety and health can be measured from four aspects: cognitive distance, emotional distance, expected distance and behavioral distance. The following conclusions were obtained from the data analysis.

Firstly, the public’s psychological distance to occupational safety and health is at a relatively close level, which indicates that the public has a high endogenous tendency to be concerned about occupational safety and health problems. On the specific dimensions, nearly half of the public investigated had a far cognitive distance with regard to occupational safety and health. The public’s distances to occupational safety and health, from near to far, are behavioral distance, emotional distance, expected distance and cognitive distance respectively. In general, the cognitive and expected distance are large, while the emotional and behavioral distance are small.

Second, a “gap” was demonstrated for the public’s cognitive, emotional, expected and behavioral distance with regard to occupational safety and health problems; the proportion of the group with a “gap” was up to 81.25%.

Third, there was a significant positive correlation between the cognitive, emotional, expected and behavioral distance for problems related to occupational safety and health, with cognitive, emotional and expected distance all able to significantly predict behavioral distance.

Fourth, further analysis showed that the consistency in the public’s cognitive and emotional distance for occupational safety and health had a progressively decreasing response to behavioral distance. That is, the higher the consistency existing in the public’s cognitive and emotional distance for occupational safety and health, the shorter the behavioral distance and the larger the gap between cognitive and emotional distance, the farther the behavioral distance. The consistency existing in the public’s cognitive and expected distance for occupational safety and health shows a “convex” change with behavioral distance. That is, when the cognitive and expected distance are consistent, the public’s behavioral distance to occupational safety and health first increases and then decreases. Generally, the consistency existing in the public’s emotional and expected distance for occupational safety and health shows a progressively increasing response to behavioral distance. However, the final trend is that the public’s behavioral distance for occupational safety and health will be shorter when the gap between emotional and expected distance is very small.

### 6.2. Policy Implications

In order to improve the public’s response to occupational safety and health, and effectively deal with, control and reduce the risk of occupational safety and health problems, the following policy recommendations are put forward according to the results of this study.

Firstly, the shortfall in the public’s psychological distances for occupational safety and health should be filled, especially to decrease the public’s cognitive distance to occupational safety and health. Cognitive distance is the basis of emotional, expected and behavioral distance, and the emotions, expectations and behaviors formed without the cognition of occupational safety and health are currently unstable. Therefore, it is necessary to improve the education of the public regarding occupational safety and health. On the one hand, the government should strengthen the popularization of knowledge related to occupational safety and health through innovative approaches to propaganda, broadening channels and intensifying publicity. On the other hand, occupational safety and health management is an activity that must be participated in by all, and the government should adopt a simple and interesting format when formulating pertinent policies or taking relevant measures, so as to maximize the public’s enthusiasm for knowledge related to occupational safety and health.

Second, the public should be guided to progress from the gap-type of psychological distance to a consistent psychological distance. From the previous analysis, we could see that the causes of the “strong attitude–weak behavior” gap were individual ability and situational dilemma. The government should strengthen the public to improve their individual abilities, eliminate constraints and create an environment for behaviors that are beneficial to occupational safety and health. With regard to “weak attitude–strong behavior”, it is known that individuals hope to express themselves and gain face through certain behaviors in view of the promotional effect of occupational safety and health governance on personal status, such as donating to patients who have suffered occupational injury. According to the theory of self-perception, people judge their own attitude to any particular occurrence on the basis of their own behavior and, to a certain extent, the situation. Based on theories of compensatory psychological mechanisms, the kind of behavior that is beneficial to occupational safety and health will also enhance the self-perception of the individual’s occupational safety and health, including cognitive, emotional and expected distance. The theory of behavioral learning level also advocates changing attitudes through behavior. It can be seen that the psychological compensation effect can not only be directly beneficial to professional safety and health behavior, but also further promote the public’s positive attitude towards occupational safety and health. This will, finally, repair the “gaps” of cognitive, emotional, expected and behavioral distance.

Thirdly, it is necessary to distinguish between the public’s behavioral distance values for various kinds of occupational safety and health. The mechanisms for forming different types of behavior distance are distinct, so that the gaps for cognitive, emotional, expected and behavioral distance are also different. Understanding these processes is a prerequisite for policy formulation, and neglect of gaps by the government and related enterprises will lead to mismatches between policy and implementation. In the future, we should further explore the process by which the public forms different types of behavior distance to occupational safety and health, and investigate the causes of the gaps generated between behavioral distance and cognitive, emotional, and expected distance. Further deepening the public’s motives for certain behaviors that are beneficial to occupational safety and health, constantly strengthening the public’s cognition of, cultivating the public’s affection for, and raising the public’s expectations of occupational safety and health will fundamentally alter the contradiction between “different knowing and doing” and promote the public’s level of response to occupational safety and health.

### 6.3. Study Limitations

This research also has some study limitations. The first one is limited research sample. Restricted by survey conditions and time, the research just recollects 2386 valid questionnaires. Though they can well represent public distribution condition, and meet basic sampling requirements under the statistical research method, the chosen samples are still limited in regional distribution. Even though the research has chosen samples from two provinces in eastern, two in middle and two in western regions, it still fails to totally reflect public occupational safety and health psychological distance conditions. Therefore, further research should enlarge the sample size. Secondly, as all data of items come from the retrospective answers of respondents, memory bias is inevitable. Thirdly, as psychological distance is perceived by individuals, corresponding theoretical construction is very sophisticated. Therefore, it is essential to further revise and modify public occupational safety and health psychological distance scale.

## Figures and Tables

**Figure 1 ijerph-16-01944-f001:**
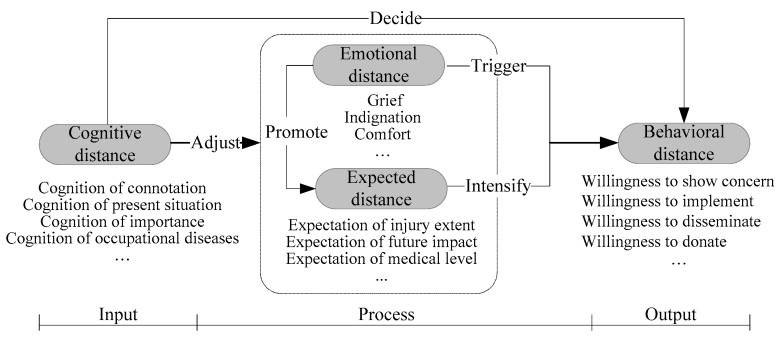
Profile chart of public psychological distance structural relationships in occupational safety and health behaviors.

**Figure 2 ijerph-16-01944-f002:**
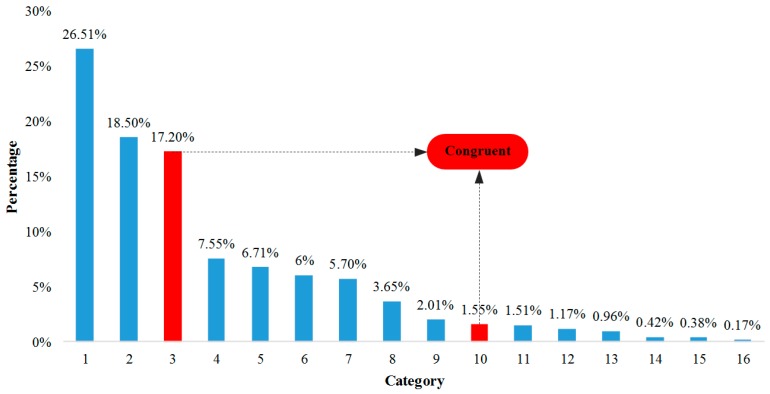
Public occupational safety and health psychological distance four-dimension gap analysis. Note: 1: cognitive distance (CD) ≥ 3, emotional distance (ED) < 3, expected distance (ExD) < 3, behavioral distance (BD) < 3; 2: CD ≥ 3, ED < 3, ExD ≥ 3, BD < 3; 3: CD < 3, ED < 3, ExD < 3, BD < 3; 4: CD ≥ 3, ED ≥ 3, ExD < 3, BD ≥ 3; 5: CD < 3, ED < 3, ExD ≥ 3, BD < 3; 6: CD ≥ 3, ED < 3, ExD ≥ 3, BD ≥ 3; 7: CD ≥ 3, ED < 3, ExD < 3, BD ≥ 3; 8: CD ≥ 3, ED ≥ 3, ExD < 3, BD < 3; 9: CD ≥ 3, ED ≥ 3, ExD ≥ 3, BD < 3; 10: CD ≥ 3, ED ≥ 3, ExD ≥ 3, BD ≥ 3; 11: CD < 3, ED < 3, ExD ≥ 3, BD ≥ 3; 12: CD < 3, ED < 3, ExD < 3, BD ≥ 3; 13: CD < 3, ED ≥ 3, ExD < 3, BD < 3; 14: CD < 3, ED ≥ 3, ExD ≥ 3, BD < 3; 15: CD < 3, ED ≥ 3, ExD < 3, BD ≥ 3; 16: CD < 3, ED ≥ 3, ExD ≥ 3, BD ≥ 3. The bold indicates congruence.

**Figure 3 ijerph-16-01944-f003:**
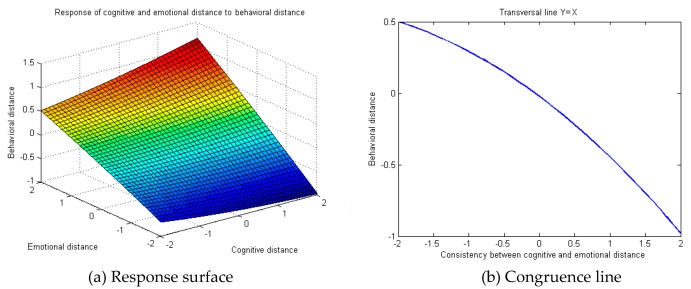
Response diagram of public cognitive and emotional distance towards occupational safety and health against behavioral distance.

**Figure 4 ijerph-16-01944-f004:**
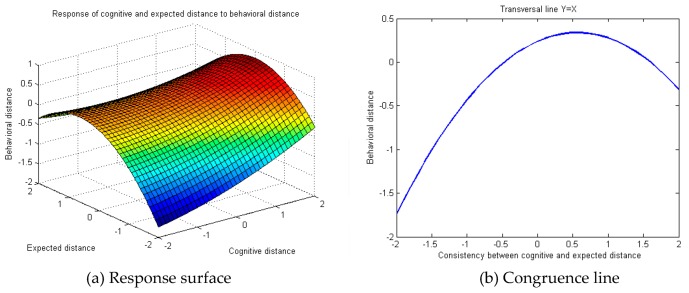
Response diagram of public cognitive and expected distance towards occupational safety and health against behavioral distance.

**Figure 5 ijerph-16-01944-f005:**
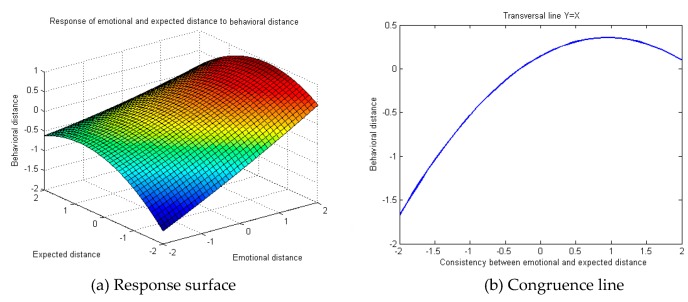
Response diagram of public emotional and expected distance towards occupational safety and health against behavioral distance.

**Table 1 ijerph-16-01944-t001:** Sample structure.

Social-Demographic Variable		Frequency (*N*)	Proportion (%)	Social-Demographic Variable		Frequency (*N*)	Proportion (%)
Gender	Male	1782	74.69	Age	<20	8	0.34
	Female	604	25.31		21–30	671	28.12
Education	Primary school and below	61	2.56		31–40	775	32.48
	Junior middle school	403	16.89		41–50	556	23.30
	Senior middle school	635	26.61		51–60	273	11.44
	Junior college	600	25.15		>60	103	4.32
	Undergraduate	469	19.66	Marital status	Single	306	12.82
	Master and higher	218	9.14		Married	1978	82.90
Identity	Government staff	109	4.57		Divorced	69	2.89
	Coal mine enterprise leader	106	4.44		Widowed	33	1.38
	Coal mine safety supervisor	155	6.50	Political status	CPC member	660	27.66
	Coal mine front-line worker	887	37.18		Democratic party	48	2.01
	Third-party social staff	210	8.80		Non-Party personage	169	7.08
	Pneumoconiosis patient	207	8.68		The mass	1509	63.24
	Ordinary people	712	29.84				

Note: CPC = Communist Party of China, *N* = number.

**Table 2 ijerph-16-01944-t002:** Descriptive statistical analysis of psychological distance (*N* = 2386).

Variable	M	SD	(1–2)	(2–3)	(3–4)	(4–5)
PD	2.5520	0.53745	454, 19.04%	1434, 60.15%	495, 20.76%	1, 0.04%
CD	3.2791	0.90552	316, 13.24%	580, 24.31%	1111, 46.56%	379, 15.88%
ED	2.1954	0.68459	1194, 50.08%	929, 38.97%	243, 10.19%	18, 0.76%
ExD	2.5624	0.96773	900, 37.72%	797, 33.40%	535, 22.42%	154, 6.45%
BD	2.1698	0.89793	1286, 53.90%	792, 33.19%	250, 10.48%	58, 2.43%

Note: PD = Psychological distance, CD = cognitive distance, ED = emotional distance, ExD = expected distance, BD = behavioral distance, *N* = number, M = mean value, SD = standard deviation.

**Table 3 ijerph-16-01944-t003:** Correlation analysis of public cognitive/emotional/expected/behavioral distance towards occupational safety and health.

Variable	Observed Value	Cognitive Distance	Emotional Distance	Expected Distance	Behavioral Distance
Cognitive distance	Pearson correlation	1			
	Sig. (2-tailed)				
Emotional distance	Pearson correlation	0.280 **	1		
	Sig. (2-tailed)	0.000			
Expected distance	Pearson correlation	0.129 *	0.05 1*	1	
	Sig. (2-tailed)	0.044	0.012		
Behavioral distance	Pearson correlation	0.180 **	0.564 **	0.079 **	1
	Sig. (2-tailed)	0.000	0.000	0.000	

Note: Sig. = significance. * indicates significant correlation at 0.05 level (bilateral). ** indicates significant correlation at 0.01 level (bilateral).

**Table 4 ijerph-16-01944-t004:** Regression analysis on public occupational safety and health psychological distance.

Variable	Behavior M1	Behavior M2	Variable	Behavior M1	Behavior M2	Variable	Behavior M1	Behavior M2
Constant	0.001	−0.024	Constant	0.000	0.239 ***	Constant	0.001	0.147 ***
Cognition	0.023	0.047 *	Cognition	0.180 ***	0.190 ***	Emotion	0.428 ***	0.359 ***
Emotion	0.425 ***	0.416 ***	Expectation	0.080 ***	0.166 ***	Expectation	0.055 **	0.084 ***
Cognition ^2^		0.011	Cognition ^2^		0.045 **	Emotion ^2^		0.016 *
Emotion ^2^		−0.007	Expectation ^2^		−0.272 ***	Expectation ^2^		−0.162 ***
Cognition × Emotion		0.058 ***	Cognition × Expectation		−0.088 ***	Emotion × Expectation		−0.088 ***
Adjustment R^2^	0.318	0.323		0.037	0.125		0.320	0.352

Note: M1: model 1, M2: model 2. R^2^: R^2^ is the ratio of the sum of the squares of the regression to the sum of the squares used to measure the goodness of fit of the model. * indicates significant correlation at 0.05 level (bilateral). ** indicates significant correlation at 0.01 level (bilateral). *** indicates significant correlation at 0.001 level (bilateral). Cognition ^2^: Cognition distance square, Emotion ^2^: Emotion distance square, Cognition × Emotion: Interaction item of cognition distance and emotion distance.

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
