# Peer review of "Chinese Public Response to Occupational Safety and Health Problems—A Study Based on Psychological Distance"

_ijerph, 2019, doi:10.3390/ijerph16111944_

Round 1

Reviewer 1 Report

This study tries to incorporate psychological distance into the study of occupational safety and health problems. The authors adopt a complex model of public psychological distance structural relationships, including four dimensions of occupational safety and health.

Sampling was accurate and the sample size should reassure that the variables have a normal distribution, which is an indispensable condition for the application of the statistics.

The results are discussed in depth and the conclusions drawn are appropriate.

Author Response

Many thanks to the reviewer for your positive recognition and full affirmation of my manuscript. We feel much honored that the reviewer made positive comments on the significance of topic selection, method adoption, sample selection, results and conclusions, which enhanced my confidence in academic research. Thanks a lot!

Reviewer 2 Report

The paper reports on survey research and consequent conceptualization of psychological distance to occupational health and safety. It is limited in geographical origin and this should be emphasized in the title, that this is specific for a large but unique country (mainland China). It flows towards policy implications, which again concern only China. Little interest is left hence for the international readership beyond China, for this paper. The manuscript needs a major overhaul to over comethese shortcomings. More detailed aspects are analysed in what follows.

What is missing in this manuscript, besides a sense of relativity, and a critical attitude towards the study design and the results, is a logical, coherent and well framed transition between theoretical and research methods sections. Why do the authors use a quadratic response surface regression analysis and what is the rationale for foregoing other methods. It is surprising that after such an extensive literature review on psychological distance, that the methods section is not an logical outcome of the literature review, as the theoretical section fails to present the methods used in previous study, and does not inform on the adequacy of the method presented without any rationale for choice. Figure 1 is at best a hypothetical model, and it should be carefully presented. Currently the one sentence paragraph is not enough, not even as a summary of the content of the Figure. The Model shown is counter intuitive, how does emotional distance trigger willingness to show concern, implement, disseminate, donate,  shouldn't it be emotional closeness?

What are the perceived advantages of quadratic response surface regression analysis over other methods commonly used (identify them and compare them). What are the limitations of using this method, again in comparison with other methods?

Moreover, convenience sampling should be declared as a limitation of the method.

Questionnaires used to collect data should be appended to the manuscript, preferably accompanied with an English translated version. Without the actual scale items in English, readers have no chance of interpreting the Cronbach alpha values given.

What is "The mass" in Table 1?

In lines 527-528, please revise the "strong attitude - weak behaviour" and "weak attitude - strong behaviour" as they seem to be swapped in order.

It is very uncanny that this manuscript does not have a large section of several paragraphs  on Study Limitations!

Author Response

Thank you very much for giving us an opportunity to revise our manuscript entitled “Public Response to Occupational Safety and Health Problems—A Study based on Psychological Distance”. The comments are valuable and very helpful for revising and improving the quality of our paper. We have studied comments carefully and have made revisions thoroughly for our manuscript in accordance with reviewers' suggestions.

We have carefully responded to all of the reviewers’ comments point by point, and submitted the original manuscript showing clearly all textual changes highlighted in red and the final-version revised manuscript respectively. The revised manuscript has been further proofread by professional English writing organizations.

Point 1: The paper reports on survey research and consequent conceptualization of psychological distance to occupational health and safety. It is limited in geographical origin and this should be emphasized in the title, that this is specific for a large but unique country (mainland China). It flows towards policy implications, which again concern only China. Little interest is left hence for the international readership beyond China, for this paper. The manuscript needs a major overhaul to over comethese shortcomings. More detailed aspects are analysed in what follows.

Response 1:

Thanks for your valuable comments. The article is really about Chinese public response to occupational safety and health. According to your valuable suggestion, the title was changed into “Chinese Public Response to Occupational Safety and Health Problems—A Study based on Psychological Distance”.

Point 2: What is missing in this manuscript, besides a sense of relativity, and a critical attitude towards the study design and the results, is a logical, coherent and well framed transition between theoretical and research methods sections. Why do the authors use a quadratic response surface regression analysis and what is the rationale for foregoing other methods. It is surprising that after such an extensive literature review on psychological distance, that the methods section is not an logical outcome of the literature review, as the theoretical section fails to present the methods used in previous study, and does not inform on the adequacy of the method presented without any rationale for choice. Figure 1 is at best a hypothetical model, and it should be carefully presented. Currently the one sentence paragraph is not enough, not even as a summary of the content of the Figure. The Model shown is counter intuitive, how does emotional distance trigger willingness to show concern, implement, disseminate, donate, shouldn't it be emotional closeness?

Response 2:

Thanks for your valuable comments. Combining with your suggestions, we elaborately supplement the logic between theoretical and research methods sections to ensure the coherence and transition of framework in the article as below (see lines 268-277).

“Based on literature review, there exists certain relation between four dimensions in public occupational safety and health psychological distance (Fig.1), but until now few scholars have studied the influence of matching degree between cognition and emotion, cognition and expectation, emotion and expectation on behaviors. Public intimacy with occupational safety and health behavioral distance should not be simply ascribed to single factor like cognitive distance, emotional distance, behavioral distance, but should be the joint effect caused by the interaction of cognitive, emotional and expected distances. Therefore, discussion about the influence of matching degree between cognitive, emotional and expected distances on behavioral distance has very important theoretical and practical values. This research plans to conduct the research with quadratic response surface regression analysis method.”

It is worth noticing here that public occupational safety and health psychological distance is our major theoretical innovation. Meantime, in combination with relevant knowledge of psychological distance, this research builds up a four-dimensional structure for occupational safety and health psychological distance, starting from the dimensions of cognitive, emotional, expected and behavioral distance to explore public attitudes and responses to occupational safety and health problems. Therefore, no pertinent method has been proposed in related studies. However, in specific matching measurement research, we carefully compare the advantages and disadvantages between quadratic response surface regression analysis method and other matching methods, the suitability of quadratic response surface regression analysis is obtained in this study, which lays a good theoretical foundation for the subsequent analysis (see lines 290-316).

“In matching measurement research, there often include two measurement methods. The first one is direct matching measurement method which requires the respondent to directly determine the degree of matching between the two [38]. Though the method is rather simple, but the independent effects cannot be examined. Another method is indirect matching measurement which requests the respondent to respectively assess the features of the two, and then compare the consistency or similarity of their grading scores [39]. Indirect measurement usually adopts difference scores, Q classification method, product term indicator method, etc. However, these indirect matching measurement statistical strategies are defective in theory and methodology. For instance, difference scores can result in decreased measurement reliability and mutual confusion between the two effects [40]. Quadratic response surface regression analysis method belongs to a matching measurement and statistical analysis strategy developed to overcome above-mentioned defects [35]. It possesses the following advantages: Firstly, it provides an entire statistical analysis and explanation framework for matching measurement research that can not only well explain the secondary coefficient of polynomial regression equation, but also test the response surface features and shape formed by these coefficients. Secondly, it is not only limited to measuring the matching relation of representatives, but also can examine the influence caused by mismatching relation, thus providing a more robust evaluation method for the theoretical model related to matching research. Thirdly, compared with the traditional indirect matching method, it has higher predictive validity. Admittedly, the method also has limitations. Firstly, as quadratic response surface regression analysis adopts a series of statistical tests to judge the slope and curvature significance of all target lines on the response surface, it greatly increases statistical mistakes due to repeated examination [35]. Secondly, quadratic response surface regression analysis has the same problem with general regression analysis application. All of these methods assume measured indicator variables have no error, but this is not up to the real measurement conditions. Though quadratic response surface regression analysis does not totally jump out of traditional statistical analysis framework, available statistical analysis technology indicates that quadratic response surface regression analysis has extensive application prospects in matching or consistency research.”

We elaborately introduce the hypothetical model (Fig.1) to fully supplement the relation among the four dimensions of public occupational safety and health psychological distance as below (see lines 250-260).

“In summary, there exists certain relation among the four dimensions of public occupational safety and health psychological distance. Public occupational safety and health cognitive distance as the basis can adjust emotional distance and expected distance, and play a decisive role in behavioral distance. Furthermore, emotional distance can promote expected distance. If the public have closer occupational safety and health emotional distance, they probably have closer expected distance and vice versa. Simultaneously, emotional distance can directly trigger behavioral distance, which means that if the public have closer occupational safety and health emotional distance, they probably show high willingness of concern, willingness of implementing, willingness of dissemination and willingness of donations to occupational safety and health problems and vice versa. Expected distance can intensify behavioral distance. If the public have closer occupational safety and health expected distance, they probably have closer behavioral distance.”

Willingness of concern, willingness of implementing, willingness of dissemination and willingness of donations in the paper are separately different in the degree of willingness. Point 1 means weakest willingness, while point 5 means strongest willingness. Likewise, grief, indignation and comfort in emotional distance are also respectively different in degree, and there exists a correspondence between the two of them.

Point 3: What are the perceived advantages of quadratic response surface regression analysis over other methods commonly used (identify them and compare them). What are the limitations of using this method, again in comparison with other methods?

Response 3:

Thanks for your valuable comments. According to your suggestion, we have supplemented in detail the advantages and disadvantages of quadratic response surface regression analysis and other methods as follows (see lines 290-316):

“In matching measurement research, there often include two measurement methods. The first one is direct matching measurement method which requires the respondent to directly determine the degree of matching between the two [38]. Though the method is rather simple, but the independent effects cannot be examined. Another method is indirect matching measurement which requests the respondent to respectively assess the features of the two, and then compare the consistency or similarity of their grading scores [39]. Indirect measurement usually adopts difference scores, Q classification method, product term indicator method, etc. However, these indirect matching measurement statistical strategies are defective in theory and methodology. For instance, difference scores can result in decreased measurement reliability and mutual confusion between the two effects [40]. Quadratic response surface regression analysis method belongs to a matching measurement and statistical analysis strategy developed to overcome above-mentioned defects [35]. It possesses the following advantages: Firstly, it provides an entire statistical analysis and explanation framework for matching measurement research that can not only well explain the secondary coefficient of polynomial regression equation, but also test the response surface features and shape formed by these coefficients. Secondly, it is not only limited to measuring the matching relation of representatives, but also can examine the influence caused by mismatching relation, thus providing a more robust evaluation method for the theoretical model related to matching research. Thirdly, compared with the traditional indirect matching method, it has higher predictive validity. Admittedly, the method also has limitations. Firstly, as quadratic response surface regression analysis adopts a series of statistical tests to judge the slope and curvature significance of all target lines on the response surface, it greatly increases statistical mistakes due to repeated examination [35]. Secondly, quadratic response surface regression analysis has the same problem with general regression analysis application. All of these methods assume measured indicator variables have no error, but this is not up to the real measurement conditions. Though quadratic response surface regression analysis does not totally jump out of traditional statistical analysis framework, available statistical analysis technology indicates that quadratic response surface regression analysis has extensive application prospects in matching or consistency research.”

Point 4: Moreover, convenience sampling should be declared as a limitation of the method.

Response 4:

Thanks for your valuable comments. We are sorry that our research method is not convenience sampling. In reality, stratified sampling is the sampling method of this research. Options for region, subject, gender, educational background, age and marital status ensure the representativeness and validity of samples. Even so, the selection of research samples is still barely satisfactory as the chosen samples can’t totally show public occupational safety and health psychological distance. Therefore, we will further expand the size of samples (see Study Limitations). Here is the specific sampling process (see lines 379-386).

“Before distributing the questionnaires, regional respondents were selected via stratified sampling. Due to the diverse economic and regional features in the eastern, middle and western regions, the research screened two cities from each of the three regions: Hebei and Jiangsu Province in the eastern region, Anhui and Hunan Province in the middle region, and Sichuan and Xinjiang Province in the western region. Simultaneously, in consideration of the wide variety of potential respondents concerned with occupational safety and health problems, including government staff, coal mine enterprise leaders, coal mine safety supervisors, coal mine front-line workers, third-party social organization staff (including those working for occupational disease hospitals, occupational disease relief funds, and industrial association organization staffs), and patients with diseases such as pneumoconiosis as well as ordinary people, seven types of respondents were selected for the investigation to provide a sample including people from all walks of life and of both genders and different educational backgrounds, ages, and marital and political status. To be specific, the research chose 3 samples from 45 state-owned large and medium coal mine enterprises in China, and selected six types of coal mine front-line workers in ventilation and fire prevention, coal mining, tunneling, mechatronics, transportation, and ground work and leaders with different positions. Moreover, the research chose coal mine safety supervisors according to age and position discrepancy. For ensuring the rationality of third-party social organization staff samples, the research chose staff and volunteers from Chinese Occupational Safety and Health Association, China Coal Miner Pneumoconiosis Prevention Foundation, and Love Save Pneumoconiosis. The sample selection conformed to the practical distribution situation. The subject and purpose of the investigation were explained in detail to the participants. The researchers sent gifts to participants to express their gratitude and to improve the response rate and effectiveness of the questionnaire survey. ”

Point 5: Questionnaires used to collect data should be appended to the manuscript, preferably accompanied with an English translated version. Without the actual scale items in English, readers have no chance of interpreting the Cronbach alpha values given.

Response 5:

Thank you for your valuable advice, and we have added the scale items for public occupational safety and health psychological distance, which can truly reflect the exact meaning of the items and dimensions (see Appendix).

Appendix: The scale items for public occupational safety and health psychological distance

Dimensions

Items   descriptions

Cognitive distance

I am very familiar with occupational safety and health   implications

I am very familiar with occupational safety and health   present situation

I am very familiar with the importance of occupational   safety and health

I am very familiar with occupational disease-related   knowledge in occupational safety and health

Emotional

distance

I feel grief about domestic security incidents and   occupational diseases in my mind

I feel indignation about the frequent occurrence condition   of domestic security incidents and occupational diseases in my mind

I feel comfort about the high attention paid by the country   and enterprises to occupational safety and health problems

I always focus on the development of occupational safety   and health in my mind

Expected

distance

I think that the public perceptions of occupational safety   accidents and the extent of injury are exaggerated

I think that the development of medical treatments and techniques   ensures that occupational safety and health problems will not severely   threaten the public

I think that the quality of occupational safety and health   problem governance has influence on the future perceptions of the public

I think that occupational safety and health problems will   be controlled within a rational scope if everyone stresses them

Behavioral distance

I am willing to concentrate on occupational safety and   health problems in response to the call of the country

I am willing to commit to occupational safety and health   problem governance

I am willing to popularize occupational safety and health   knowledge to surrounding people

I am willing to make donations to those injured in   occupational accidents and occupational disease patients

Point 6: What is "The mass" in Table 1?

Response 6:

Thank you for your valuable advice. “The mass” refers to “common people”.

Point 7: In lines 527-528, please revise the "strong attitude - weak behaviour" and "weak attitude - strong behaviour" as they seem to be swapped in order.

Response 7:

Thank you for your careful work. We are so sorry for our vague expression. We have switched the order of "strong attitude - weak behaviour" and "weak attitude - strong behaviour".

Point 8: It is very uncanny that this manuscript does not have a large section of several paragraphs on Study Limitations!

Response 8:

Thanks for your valuable comments. According to your valuable suggestion, we add one chapter about “Study Limitations” (see lines 729-740).

“Study Limitations

This research also has some study limitations. The first one is limited research sample. Restricted by survey conditions and time, the research just recollects 2386 valid questionnaires. Though they can well represent public distribution condition, and meet basic sampling requirements under the statistical research method, the chosen samples are still limited in regional distribution. Even though the research has chosen samples from two provinces in eastern, two in middle and two in western regions, it still fails to totally reflect public occupational safety and health psychological distance conditions. Therefore, further research should enlarge the sample size. Secondly, as all data of items come from the retrospective answers of respondents, memory bias is inevitable. Thirdly, as psychological distance is perceived by individuals, corresponding theoretical construction is very sophisticated. Therefore, it is essential to further revise and modify public occupational safety and health psychological distance scale.”

Reviewer 3 Report

This is an important topic, and a novel and interesting study, especially for developing countries.

The definition of psychological distance in the earlier section could be clearer, as could the 4 further areas within it. The information on p.3 and 4 is much clearer, and the ideas could be brought together better overall. Tangible examples of what each one means for lay readers unfamiliar with these concepts would be useful.

I cannot comment on the stats as this is not my area of expertise.

The manuscript would benefit from some further input by an English editor. The concepts are complex enough as it is, so this would be important to ensure the wider lay research readership understands the concepts and their implications in tangible terms. For example, the term ‘public affairs’ may not be familiar to many. Framing occupational safety and health as a public health concern may be more widely understood.

Author Response

Many thanks to the reviewer for your positive recognition and full affirmation of my manuscript. We feel much honored that the reviewer made positive comments on the significance of topic selection. The comments are valuable and very helpful for revising and improving the quality of our paper. We have studied comments carefully and have made revisions thoroughly for our manuscript in accordance with reviewers' suggestions.

Point 1: This is an important topic, and a novel and interesting study, especially for developing countries.

Response 1:

Many thanks for your positive recognition.

Point 2: The definition of psychological distance in the earlier section could be clearer, as could the 4 further areas within it. The information on p.3 and 4 is much clearer, and the ideas could be brought together better overall. Tangible examples of what each one means for lay readers unfamiliar with these concepts would be useful.

Response 2:

Thanks for your valuable comments. According to your valuable suggestion, we have added the definition of psychological distance and four dimensions. We have also added the tangible examples (see lines 64-66 and lines 136-141).

“Agnew et al. (2004) defined psychological distance as “individuals’ subjective perception about others and self-relation distance and resulting emotional experience after integrating all sorts of social information”. Spatial distance is defined as the distance between stimulant and individual in spatial dimension; temporal distance is defined as the time between individual and present and target events in the past or future; social distance is defined as the intimacy or specificity of the relation between social objects and individual; hypotheticality is defined as the occurrence possibility of event, existence possibility of thing, or closeness degree to individual real life.”

In addition, we also list some tangible examples to help readers better understand the meaning of these concepts as below (see lines 143-149).

“In their study of employee-organization psychological distance, Chen and Li (2018) grouped psychological distance into six dimensions: experiential, behavioral, emotional, cognitive, time-space and objective social distance. For example, if employees felt close to organization time-space distance, they would like to stay in the organization (voluntary overtime). Otherwise, they might get out of the organization immediately after work. If employees felt close to organization emotional distance, they would have sense of happiness in the organization or otherwise suffer from pains and depression. If employees felt close to organization behavioral distance, they might sacrifice their own interests to safeguard the interests of the organization. On the contrary, they would pursue their own best interests in the organization.”

Point 3: The manuscript would benefit from some further input by an English editor. The concepts are complex enough as it is, so this would be important to ensure the wider lay research readership understands the concepts and their implications in tangible terms. For example, the term ‘public affairs’ may not be familiar to many. Framing occupational safety and health as a public health concern may be more widely understood.

Response 3:

Thank you for your careful work. According to your valuable suggestion, we have changed "public affairs" to "public health concern". We have consulted a professional language editing service to check the English. We thank International Science Editing (http://www.internationalscienceediting.com) for editing this manuscript.

Round 2

Reviewer 2 Report

Please have the English carefully edited. Also notice that security and safety have different meanings. Revise very carefully!

Reviewer 3 Report

Minor grammatical issues remains, please revise them when proof reading.